# Deep reefs of the Great Barrier Reef offer limited thermal refuge during mass coral bleaching

Pedro R. Frade [1], Pim Bongaerts[2,3,4], Norbert Englebert[2,3,5], Alice Rogers[5,6], Manuel Gonzalez-Rivero [2,3,7] & Ove Hoegh-Guldberg[2,3]

Our rapidly warming climate is threatening coral reefs as thermal anomalies trigger mass coral bleaching events. Deep (or "mesophotic") coral reefs are hypothesised to act as major ecological refuges from mass bleaching, but empirical assessments are limited. We evaluated the potential of mesophotic reefs within the Great Barrier Reef (GBR) and adjacent Coral Sea to act as thermal refuges by characterising long-term temperature conditions and assessing impacts during the 2016 mass bleaching event. We found that summer upwelling initially provided thermal relief at upper mesophotic depths (40 m), but then subsided resulting in anomalously warm temperatures even at depth. Bleaching impacts on the deep reefs were severe (40% bleached and 6% dead colonies at 40 m) but significantly lower than at shallower depths (60–69% bleached and 8–12% dead at 5-25 m). While we confirm that deep reefs can offer refuge from thermal stress, we highlight important caveats in terms of the transient nature of the protection and their limited ability to provide broad ecological refuge.

[1] Centre of Marine Sciences, University of Algarve, 8005-139 Faro, Portugal. [2] Global Change Institute, The University of Queensland, St Lucia, QLD 4072, Australia. [3] Australian Research Council Centre of Excellence for Coral Reef Studies, The University of Queensland, St Lucia, QLD 4072, Australia. [4] California Academy of Sciences, San Francisco, CA 94118, USA. [5] School of Biological Sciences, The University of Queensland, St Lucia, QLD 4072, Australia. [6] School of Biological Sciences, Victoria University of Wellington, Wellington 6140, New Zealand. [7] Australian Institute of Marine Science, PMB 3, Townsville MC, QLD 4810, Australia. These authors contributed equally: Pim Bongaerts, Norbert Englebert. Correspondence and requests for materials should be addressed to P.R.F. (email: prfrade@ualg.pt)

Coral reefs are under immediate threat because of a rapidly changing climate and resulting episodes of excessive thermal warming that trigger coral bleaching and mass mortality[1–4]. Our understanding of the drivers of coral bleaching, mostly gathered from large-scale ecological monitoring and satellite imaging, is focused almost exclusively on shallow reef communities (~0–15 m depth)[3,5–7]. The potential for deeper sections of coral reefs, including mesophotic coral ecosystems (>30–40 m depth), to provide a refuge against thermal bleaching[8–10] remains poorly evaluated. To a large extent, this is because of the increased logistical complexity of studying these ecosystems, with only a small number of studies documenting warm-water bleaching impacts at mesophotic depths[11–14]. However, as the condition of the world's coral reefs is rapidly deteriorating, there is an urgent need to clarify the potential role of deep coral reefs to act as (short-term) refuges or (long-term) refugia from climate change[15,16], and to facilitate the recovery of shallow reefs[10].

Record-breaking sea surface temperatures brought the worst-ever recorded mass coral bleaching in the Great Barrier Reef (GBR) Marine Park in the austral summer of 2015–2016. This led to an estimated 29% bleaching-related mortality of all shallow-water corals in the GBR, with >75% of this coral loss taking place in the far northern section[3,17]. Although physical oceanographic processes are relatively well-studied in the GBR and Western Coral Sea (WCS)[18–20], long-term temperature measurements from mesophotic reef environments remain virtually absent. Concurrently, the impact of mass bleaching events (including 1998, 2002 and 2016) at mesophotic depths of the GBR remains completely unknown. Despite this lack of knowledge, mesophotic reefs are estimated to represent a surface area equivalent to that of shallow reefs on the GBR[21], and constitute >80% of the depth range at which zooxanthellate coral communities occur[22]. Here, we address the potential for the deep reefs to provide thermal refuge in this region (Fig. 1a). We do this by monitoring the long-term temperature conditions at shallow and mesophotic sites (10–100 m depth; GBR and WCS), and describing bleaching incidence down to 40 m depth during the 2016 mass-bleaching event (5–40 m depth; GBR).

## Results

**Temperature measurements**. Long-term temperature measurements revealed that thermal regimes at mesophotic depths (40, 60, 80 and 100 m depth) of the GBR and the adjacent WCS are clearly distinct from those experienced in shallow water (10 m), but only during the warmer period of the year (i.e., around the austral summer months; Fig. 2). Instead, during the colder months of June–September, the water column appears to be well mixed, with similar temperatures recorded throughout the entire depth range (e.g. <0.1 °C difference between 10 and 60 m and <0.7 °C difference between 10 and 100 m depth at all sites during 2013 winter months). During the warmer period between October–May, cold-water influxes resulted in highly variable temperatures at mesophotic depths (≥40 m), with this variability (frequency and range) becoming more pronounced with increasing depth (see Fig. 2 and Supplementary Figure 1). In the WCS, the average difference in monthly temperature between shallow (10 m) and lower mesophotic (60 m) reefs during that same period was always ≥2 °C, at least twice as much as the one registered for the GBR sites (≤1 °C). This difference is likely due to tidal mixing with heated waters from the GBR lagoon on outer reef sites[18,19], compared with the isolated oceanic setting of the WCS atolls. Shallow reefs (10 m) on the GBR sites experienced much longer periods of exposure to temperatures above 28 °C (>50% of the time during October–May in 2013–2014) as compared with those at lower mesophotic depths (<31% of the time for 60–100 m), or even at the upper mesophotic (29–46% of the time for 40 m). These differential long-term patterns of exposure to higher and lower temperatures may have resulted in different thermal stress thresholds for corals living at shallow versus mesophotic depths (as physiological tolerances are often adjusted to prevailing conditions)[12,23]. In the WCS, the periods of exposure to temperatures above 28 °C at shallower depths during the austral summer 2013–2014 were similar to those on the GBR (43–57% of the time at 10 m), but shorter at lower mesophotic depths (17–32% at 40 m and only 0.2–4% of the time at 60 m depth). Average absolute deviations from the mean daily temperature are within the same range for both the GBR and WCS sites (Supplementary Figure 1), however, weekly–monthly temperature oscillations at mesophotic depths appear more irregular on the GBR, likely the result of intermittent seasonal upwelling.

Temperature records during the 2016 mass bleaching event were anomalously high as measured in situ across locations and depths, and kept rising throughout the austral summer, peaking between the end of February and the end of March 2016 (Fig. 2). The highest registered daily averages ranged from 30.1 to 30.6 °C at 10 m and 29.7–29.9 °C at 40 m depth for the monitored GBR and WCS sites, corresponding to 1.5 and 1.8 °C above long-term averages (of the warmest month) for 10 and 40 m depth, respectively. These high late summer temperatures at 40 m depth only decreased to 28 °C by mid-April, a significant change given that temperatures were well below 28 °C already by the end of February in a typical year, according to our 3-year record. This temperature anomaly, led to cumulative exposures that greatly exceeded the average exposure recorded for the previous 3 years at 40 m depth (Fig. 3). This, despite the upwelling season being longer that summer, extending well into the fourth week of March 2016 for the GBR sites, whereas in previous years upwelling peaks always subsided before the end of the second week of March (see Fig. 2). Even though this anomalously long upwelling season brought about cold-water influxes at 40 m depth, temperatures at the upper mesophotic reef rose up to record-high levels by the time the shallow versus deep water mixing finally took place towards the end of March (Fig. 3). Maximum absolute temperatures at this depth at the GBR sites were 30.2–30.4 °C by then; quite similar to the maxima of 30.6–30.7 °C registered at 10 m depth. The average monthly temperatures at the two GBR sites were 1.3–1.6 °C higher in March 2016 (irrespective of depth) as compared with the homologous average of the previous 3 years; then, in April 2016 they were 0.9–1.1 °C higher and in May 2016 they exceeded by 1.4–1.5 °C the previous 3-year average. For the WCS these differences were even larger, with monthly temperatures during March, April and May 2016 being, respectively, 1.7–1.9 °C, 1.4–1.9 °C and 1.4 °C higher than the average of the previous years. It is remarkable to note that only when comparing the period December–May (instead of December–March) do the 2016 cumulative exposure curves at 40 m depth for all 3 sites surpass the 3-year curves for 10 m depth, revealing the relevance of the late-summer warming to the upper mesophotic reef. This delayed warming added another 3–4 °C-weeks (degree heating weeks or DHW)[24] of thermal exposure to the deep GBR sites, with an increase in DHW from 6–7 up to 9–10 °C-weeks in just 2 weeks (Fig. 3). These maximum DHW values at 40 m were identical to those registered at 10 m depth. Although the upwelling season in the WCS usually lasts much longer (i.e., until June) than on the outer GBR, it did not result in thermal relief during the 2016 bleaching. In fact, Osprey Reef was characterised by even warmer temperatures compared with the GBR, reaching absolute values of 30.6 °C and 31.0 °C by early March 2016 at, respectively, 40 and 10 m depth. DHW values at

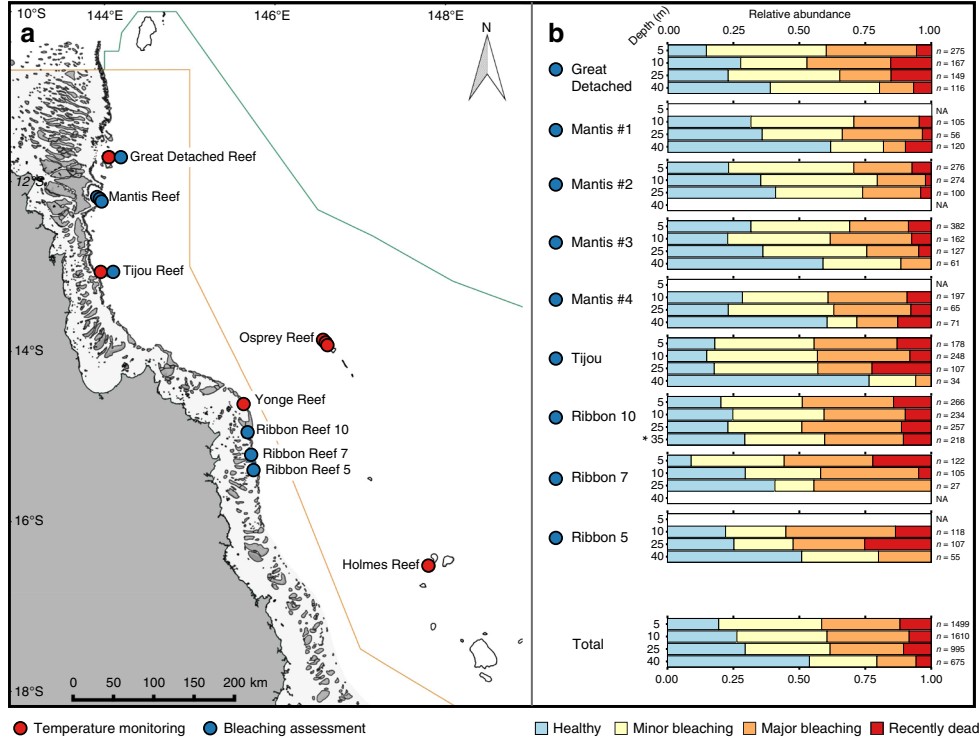

**Fig. 1** Overview of surveyed locations and respective bleaching impacts over depth. **a** Reef locations for which long-term temperature recordings (red dots) and/or bleaching surveys (blue dots) were obtained on the Great Barrier Reef (GBR) and in the Western Coral Sea (WCS). **b** Community-wide bleaching severity over depth (5, 10, 25 and 40 m depth) during the 2016 mass bleaching event. Stacked bar graphs give the relative abundance of bleaching categories for each depth and location. Number of observations noted next to each stacked bar, with NA denoting depths that lacked a coral reef community. Asterisk (*) indicates site where data was collected at 35 m depth due to absence of reef formation at 40 m. Bleaching surveys were performed on 14–23 May 2016, ~10 weeks after the first reports of minor to moderate coral bleaching in three management areas of the GBR Marine Park[17]. Map created with data files courtesy of Great Barrier Reef Marine Park Authority and www.deepreef.org under CC BY 4.0

Osprey Reef reached 10 and 13 °C-weeks in mid-April at 10 and 40 m depth, respectively. These warmer temperatures could be facilitated through the warming and tidal flushing of Osprey Reef's shallow lagoon. Regardless, our Osprey temperature data demonstrates that oceanographic settings with stronger upwelling patterns, as observed for the WCS atolls, may not necessarily convey greater thermal relief at mesophotic depths, although the actual bleaching impact was not assessed here.

**Bleaching surveys**. The impact of thermal stress (i.e. coral bleaching and mortality) was assessed during mid-May 2016, through video transects at shallow (5 and 10 m), intermediate (25 m) and upper mesophotic (40 m) depths for six reef locations across the severely bleaching-impacted northern GBR (Fig. 1a; the WCS was unfortunately inaccessible during these surveys due to bad weather). Bleaching impact (Fig. 1b) was considerably higher at shallower compared with mesophotic depths, with 69%, 65% and 60% of all colonies found to be bleached at 5, 10 and 25 m, respectively, vs. 40% at 40 m (results averaged across sites). Recent mortality (bleaching-related) affected 5.7% of the colonies at 40 m as compared with 10.5%, 8.4% and 11.8% at 25, 10 and 5 m depth, respectively. However, depth was also identified as a strong predictor of coral community composition (see Supplementary Figures 2a and 3), indicating that some of the apparent relief in bleaching incidence offered by the deep reef may actually be explained by differences in the abundance distribution of bleaching-susceptible versus bleaching-tolerant coral taxa (see Fig. 4). In order to disentangle the contributions of coral community structure and depth to the bleaching impact, we used an ordinal

logistic regression (OLR) model. By predicting the bleaching response of individual colonies ($n = 3394$), we could ascertain the separate effects of depth and taxonomic affiliation, as well as the amounts of variation explained by each of these significant factors (see Fig. 5). OLR confirmed that with increasing depth, the chance of bleaching and bleaching-related mortality decreased slightly though significantly. For a one-unit increase in depth, one should expect about 4.8% decrease (i.e., 0.952 increase; see odds ratio for depth in Fig. 5a) in the odds of being in a higher bleaching category (if holding region and coral taxa at a fixed value). Thus, the odds of progression from lower to higher bleaching categories were reduced by 39% for every 10 m of depth. This corresponds to 2.7 and 5.6 times higher odds of finding bleaching progression at 5 m than at 25 and 40 m, respectively (Fig. 5 and Supplementary Figure 4). These significant increases towards the shallow reef are supported by data collected during the same global mass-bleaching event at the Maldives[13], which showed a 2.5 times increase in the likelihood of individual corals at 3–5 m depth to experience moderate or severe bleaching as compared with those at 24–30 m. Odds of bleaching occurring and its progression to coral mortality were 2.5 times as high for coral in the Ribbon reefs as compared with the Far Northern GBR sites (see odds ratio for region in Fig. 5a), a result that is consistent with the geographical variation of the mass bleaching published recently[3]. Also, note a more acute reduction in bleaching impact with depth in the Ribbon reefs (5.6% reduction per metre in the odds of being in a higher bleaching category) than in the Far Northern GBR (4.3% reduction per metre), reflected in the significant interaction of depth and region identified in the OLR model (Fig. 5b).

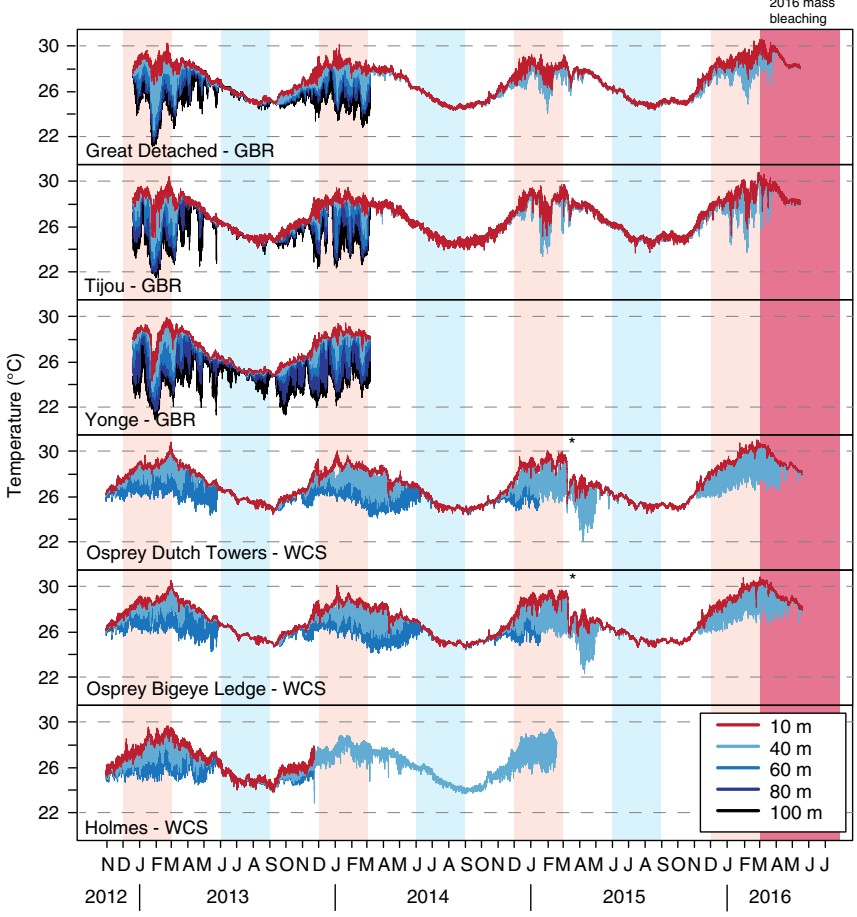

**Fig. 2** Long-term temperature recordings. Multi-annual seawater temperatures at 10–100 m depth at the Great Barrier Reef (GBR: top three sites, 10–100 m) and Western Coral Sea (WCS: bottom three sites, 10–60 m) study sites, before and during the 2016 mass bleaching event (pink shaded area). Note missing temperature records for some site–depth combinations, but continuous records for 10 and 40 m depth for two GBR and WCS sites each. On the GBR sites all recordings for 60–100 m ceased in March 2014. At the Osprey sites in the WCS the 60 m recordings ceased in January 2015, and for Holmes the 10 and 60 m loggers stopped recording in November 2013, whereas the 40 m logger went on recording until February 2015. Note sudden drop in temperature at the Osprey sites in March 2015 due to cyclone Nathan (marked by *). Bleaching surveys were performed for the GBR sites on 14–23 May 2016, coinciding with the end of the temperature records (due to data logger collection)

At our GBR sites, coral taxa *Porites*, *Leptoseris*, *Acropora* and *Pocillopora* showed a comparatively low chance of experiencing bleaching and mortality, whereas taxa such as *Stylophora*, *Isopora* and *Montipora* were particularly prone to bleaching (see Fig. 4, genus-specific odds ratios in Fig. 5a, and Supplementary Figure 4). On top of that, some bleaching-sensitive taxa seem to experience a strong effect of depth on their bleaching response (i.e., *Pachyseris*, *Dipsastraea* and *Seriatopora*), as confirmed by the odds ratios for the interaction of specific taxa with depth (Fig. 5b). Identification of "bleaching-tolerant" taxa, "bleaching-sensitive" taxa and highly sensitive taxa with "dampened response over depth" clearly reflect known taxa-specific patterns of bleaching susceptibility versus tolerance often attributed to physiological properties inherent to the coral animal itself or to its symbiotic communities[8,13,25]. The observed greater proportions of thermally tolerant coral taxa present on the shallow reef and that of highly sensitive genera on the deeper reef (Supplementary Figure 3b), suggest that the relief in bleaching incidence offered by the deep reef cannot be solely explained by differences in community composition (e.g. a more thermally tolerant community on the deep reef) and that there is at least some degree of thermal relief with depth. Our overall results on the susceptibility of coral taxa to bleaching are also well aligned with those of recent studies of coral lineages at low and high risk of bleaching-

associated extinction[13,26]. These genus-specific differences in bleaching susceptibility highlight that although the bleaching effect was quite thorough across all coral genera, there will be winners and losers as a result of long-term impacts of mass bleaching events on coral reefs, and the GBR in particular[27]. The differential susceptibility of distinct coral species to thermal stress will determine their future abundances and distribution and therefore the resulting community structure over depth[13]. Based on our OLR results and the expected increase in the intensity and frequency of occurrence of mass bleaching events[2,28], we tentatively predict a further reduction in the abundance of bleaching sensitive genera, such as *Stylophora* or *Seriatopora*[29] on the shallow reefs of the GBR, and a consistently growing contribution to the overall reef community of coral taxa that have a large proportion of their population in the upper mesophotic reef.

## Discussion

The mass bleaching event of 2016 has clearly demonstrated that, on top of the widespread coral mortality caused by a rapidly warming global climate[3], the fate of specific areas of the GBR and WCS is controlled by local oceanographic conditions[30]. Reefs along the shelf edge of the northern GBR are influenced by the

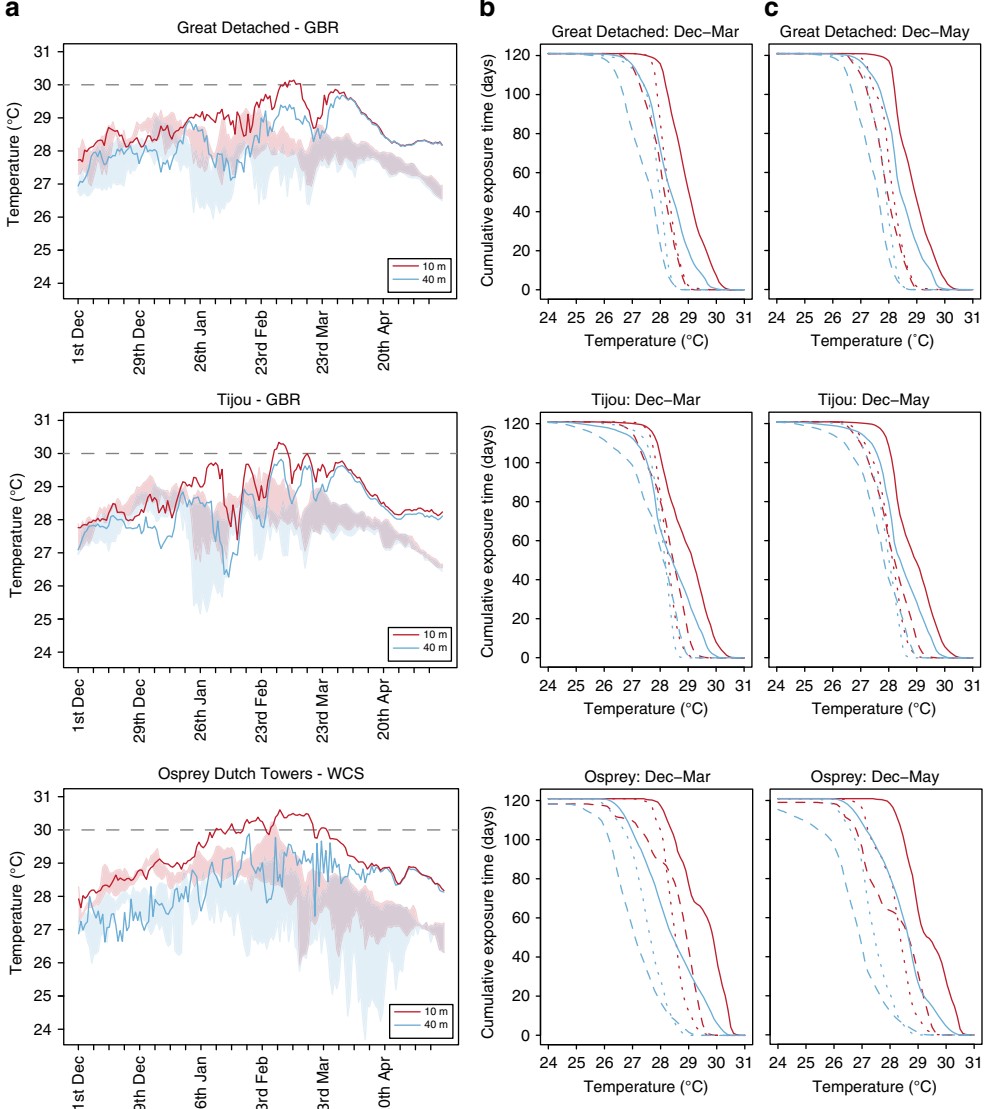

**Fig. 3** Temperature profiles during the 2016 bleaching event compared with the three previous years. **a** Daily average temperatures at 10 and 40 m during the warmest months in 2016, compared with the daily minimum and maximum temperatures recorded in the three previous years (indicated with shading). **b** Cumulative exposure times (days) and temperatures (0.1 °C) shown for three consecutive 1 December–31 March periods. **c** Cumulative exposure times (days) and temperatures (0.1 °C) shown for three consecutive 1 December–17 May periods. Dotted lines refer to 2013–2014, dashed lines to 2014–2015 and solid lines to 2015–2016

bifurcation of the South Equatorial Current as it flows westward through the Coral Sea and hits the Australian continental shelf. Its northern arm, the North Queensland Current or Hiri Current, drives along-shelf flows on the continental shelf and controls the position of the thermocline and the possibility of cooler deep water masses to access the continental shelf[18,31]. Our long-term data demonstrate the importance of such sub-thermocline upwelling[32,33] on the temperature regime of mesophotic coral ecosystems as compared with that of shallow reefs (Fig. 2). The occurrence of cold-water influxes during the summer months can provide thermal relief to mesophotic reefs, although seasonality puts a temporal constraint on this capacity when elevated temperatures extend into the winter period, as was the case in 2016. Leading up to the 2016 mass-bleaching event, the shutdown of the Hiri Current and the advection of warm water from the Gulf of Carpentaria both decisively contributed to warming the northern GBR continental shelf[30], bringing abnormally high

temperatures which subsequently drove the widespread mass bleaching of the northern GBR[3].

Our study demonstrates the ability of deep reefs, which represent a substantial proportion of overall reef habitat on the GBR and adjacent WCS[21,34,35], to act as thermal refuges during a mass bleaching event. Nonetheless, we also identified two important caveats that surround their capacity to act as refuges, despite the lower bleaching incidence at depth in 2016. First of all, the cold-water influxes responsible for providing thermal relief at mesophotic depths represent seasonal oceanographic processes, and are clearly temporally constrained. Consequently, deep reefs may not be protected from thermal anomalies when those conditions extend to later in the year, as was the case for the 2016 mass bleaching event. Due to the lag in the onset of elevated temperatures at mesophotic depths, it is possible that this bleaching event had a delayed impact on the deep reef, and that the eventual bleaching impacts were worse than measured at the

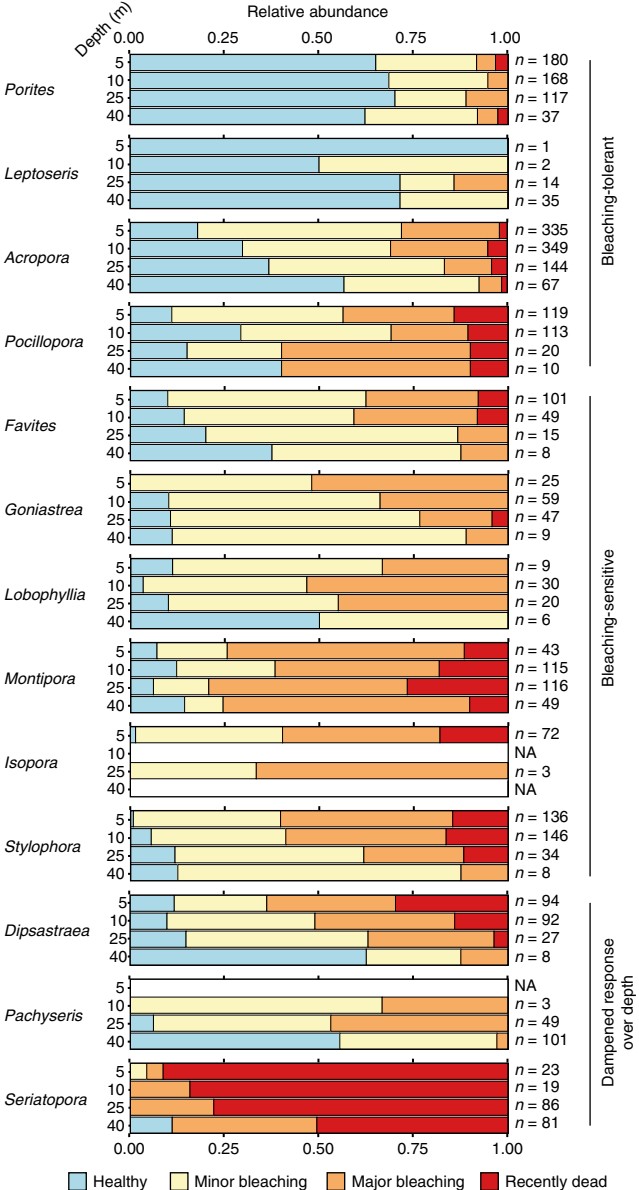

**Fig. 4** Bleaching sensitivity over depth for the most abundant coral genera with all surveyed locations combined. Great Barrier Reef only. Stacked bar graphs give relative abundance of each bleaching category within each depth population. Note the occurrence of "bleaching-tolerant" taxa (i.e., *Porites*, *Leptoseris*, *Acropora* and *Pocillopora*), "bleaching-sensitive" taxa (e.g., *Stylophora*, *Isopora* and *Montipora*) and highly sensitive taxa to bleaching with "dampened response over depth" (i.e., *Pachyseris*, *Dipsastraea* and *Seriatopora*), as sustained by our OLR bleaching model (see Fig. 5 and Supplementary Figure 4). Number of observations noted next to each stacked bar are a valid representation of the depth effect on community composition (shown in Supplementary Figure 3). NA denotes depths at which no individuals were found of that particular taxon

time of our surveys. Such a delayed onset and longer duration of heat stress in mesophotic reefs than in their shallow water counterparts has previously been shown for the Caribbean region[12]. Secondly, the lower incidence of bleaching that was observed on the deep reef coincides with a considerable shift in the community structure. This indicates that the proportion of shallow-water diversity protected in sufficiently large numbers at depth may be limited, and that a subsequent role of the meso-photic reefs as reproductive source aiding in shallow reef recovery

may be restricted to a relatively small proportion of species[36]. Although the role of deep reefs as thermal refuges remains a hopeful prospect in helping preserve coral reefs and their diversity, its broader effectiveness may be limited as warming continues.

## Methods

**Temperature measurements.** Temperature loggers (Onset HOBO Pro v2 model U22-001; Onset Computer Corp., Bourne, USA) were deployed on the ocean-facing reef slopes of three outer reef locations on the northern GBR and at two locations (four sites) in the WCS to examine the temperature regimes over a large depth gradient at each site (Fig. 1a). Temperature loggers were cable-tied to the reef substrate using SCUBA at 10 and 40 m depth. For 60, 80 and 100 m depth, temperature loggers were attached to a small PVC cylinder (containing a coated dive-weight and a short rope with two fishing net floats) and deployed using a remotely operated vehicle (SeaBotix vLBV-300). Temperature recordings were obtained for northern GBR sites, such as Great Detached Reef, Tijou Reef and Yonge Reef at depths of 10, 40, 60, 80 and 100 m for a period of ~14.5 months (20 December 2012–7 March 2014). Continuous temperature recordings, which include the period leading up and during the 2016 bleaching event, were obtained from Great Detached and Tijou at 10 and 40 m depth (till 17 May 2016), totalling a period of ~41 months. In the WCS, temperature data were obtained from Holmes Reef and three sites at Osprey Reef ("Bigeye Ledge", "Dutch Towers" and "Nautilus Wall") at 10, 40 and 60 m depth. Loggers recorded for a ~27.5-month period at Holmes (30 October 2012–16 February 2015; but note discontinuity for 10 and 60 m loggers) and for ~42.5 months at the Osprey sites (30 October 2012–17 May 2016; note discontinuity for 60 m logger). Temperature loggers recorded in situ water temperature with a 15-min interval. The initial deployed temperature loggers were exchanged for new loggers at all sites, except at Yonge Reef, during the recording period to prevent exceeding their storage capacity. Pre-deployment logger calibrations were performed at the Australian Institute of Marine Science (Townsville, Australia) and derived correction factors were applied to the data.

Cumulative exposure times and temperatures were calculated as previously published[37] to demonstrate how temperature exposures, at 10 and 40 m depth during the 2016 bleaching year, compare with the previous non-bleaching years. These cumulative exposure times and temperatures were assessed for three consecutive 1 December–31 March periods (normally the period when the warmest average daily temperatures are recorded on the GBR[37]) and 1 December–17 May periods (extended period until the bleaching surveys, conducted at Great Detached, Tijou and Osprey Reef in 2016).

Accumulated heat stress throughout the bleaching event was determined for 10 and 40 m depth using the DHW metric[24]. DHW were calculated as the accumulated weekly temperatures exceeding the average temperature of the warmest month (of the previous 3 years) using site-specific and depth-specific logger-recorded temperature data. This approach allows estimating heat stress thresholds for corals that live at mesophotic depths and whose acclimatisation history has not been considered so far. This was done by assuming that their bleaching response to temperature is based on the same premises as is well-established for shallow-water corals[3,24]. Averages of the warmest month over the period 2013–2015 were, for 10 m depth, 28.5 °C, 28.6 °C and 29.0 °C, for Great Detached, Tijou and Osprey Reef, respectively, and, for 40 m, 27.9 °C, 28.1 °C and 28.0 °C, respectively.

**Bleaching surveys.** Bleaching surveys were done between 14 and 23 May 2016, ~10 weeks after the first reports of minor to moderate coral bleaching in three management areas of the GBR Marine Park[17], with severe bleaching and high mortality confirmed soon afterwards for the far northern section of the GBR. Our surveys were performed at six different outer reefs (nine sites in total) across the northern section of the GBR between 5 and 40 m depth to determine the bleaching severity over a large depth range. Six of these sites were located on the outside of the GBR lagoon in the far northern section, whereas the other three sites were located on the inside (and were surveyed as weather conditions prevented further surveys on the seaward side) of the Ribbon reefs (Fig. 1a). Our bleaching surveys were originally intended to also cover Yonge Reef and the Coral Sea atolls, but weather was too rough to access these sites at the time of our surveys. Benthic video transects (see Supplementary Figure 5) were carried out by SCUBA divers using a high-definition GoPro camera (4K resolution) mounted on a frame with two video lights (Keldan Video Luna 8 FLUX 6000 lumen; KELDAN GmbH, Brügg Switzerland) and folding-out reference poles 1 m apart. The 1 m-wide belt transects were conducted at 5, 10, 25 and 40 m depth and each transect covered 75 m in length. Within the transects, the zooxanthellate scleractinian corals larger than 10 cm in diameter were identified to genus level and the level of bleaching severity or mortality of these corals was recorded. The bleaching impact was originally scored in six categories, however for further analyses the following four categories were utilised as these were more appropriate to reduce scoring error[38]: "1"—healthy; "2"—minor bleaching (paling/1–50% bleached); "3"—severe bleaching (51–100% bleached); and "4"—recently dead. The geographic variation in relative incidence of bleaching and bleaching-related mortality that we measured at the shallowest sampled depth (5 m) was consistent with measurements made in aerial surveys by

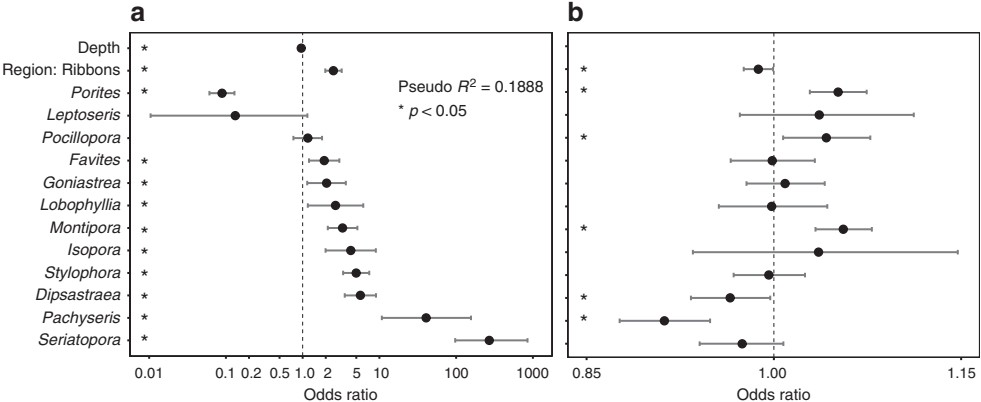

**Fig. 5** Odds ratios for the progression of bleaching categories from healthy to bleaching-related mortality. Odds ratios (with 95% confidence intervals) produced by the ordinal logistic regression model for (**a**) significant explanatory variables (depth, region and the different coral taxa), and (**b**) their interactions with depth (significant for region and coral taxa). OLR is an extension of binomial logistic regression in which multiple categories associated with an ordinal event are attributed an odds ratio statistic—the probability of the occurrence of that event based on a combination of one or more independent variables. Odds ratios for region refer to Far Northern GBR as the reference (combining Great Detached and Tijou) and for coral taxa to *Acropora* (corresponding to an odds ratio of one). An asterisk (*) marks categories or interactions for which log odds (and ratios) are significantly different from those of reference categories. Note the very narrow confidence intervals for depth in panel **a**, which do not overlap with the unit of odds ratio, therefore confirming the statistical significance of this variable. Location had no significant effect on the progression of bleaching as tested by OLR and individual locations were therefore grouped into two regions. The full model, including interaction effects between depth and each of the other two significant variables, region and coral taxa, explained 18.9% of the variation in the dataset. Depth, coral taxa and region explained 1.4%, 17.3% and 1.1% of the variation in bleaching response, respectively. The interactions depicted in panel **b** mean that the different coral genera do not respond homogeneously over depth, and that the effect of depth is not homogeneous between regions. See Supplementary Figure 4 for further modelling results

Hughes et al.[3], with Mantis Reef being the least affected of the studied locations (30% of bleached colonies in our survey vs. 1–10% in Hughes et al.[3]), followed by Great Detached Reef (40% vs. 10–30%), Tijou (44% vs. 30–60%), and finally the Ribbon Reefs (51% vs. >60%). This confirms that our study, while using distinct methods and targeting different areas of the same reefs 1–2 months later, captures similar measurements across the bleaching impact map reported by Hughes et al.[3] in March–April 2016[3].

**Statistical analyses**. Bleaching severity across depths for the GBR sites was analysed using two distinct statistical approaches: firstly by testing community-level bleaching responses with a combination of multivariate tools to analyse the abundance distribution of the distinct bleaching categories (healthy, minor bleaching, severe bleaching and recently dead) at the ecosystem level, as a response of depth, region (Far Northern sites versus Ribbon reefs) and coral taxon (included as supplementary material). Secondly, by applying a univariate approach that models the bleaching response of individual colonies and ascertains a level of predictability according to significant factors (reported in the main text). These two approaches were applied on a representative portion of the data containing only commonly occurring coral taxa (i.e., those totalling more than 50 individual bleaching scoring observations).

Multivariate statistical analyses were applied (a) at the coral community level, for the different locations sampled (all host species combined, for a total of 31 location × depth observations); and (b) between the different resolved coral taxa (all locations combined, for a total of 52 taxa × depth observations). The effect of depth, location, region and coral taxonomy on the overall distribution of distinct bleaching categories (healthy, minor bleaching, severe bleaching and recently dead) was analysed using canonical correspondence analysis (CCA) and an ANOVA-like permutation test (9999 permutations) to investigate the significance of explanatory factors selected by an automatic backward and forward model selection tool ("ordistep", based on the AIC index, with a maximum of 200 permutations). Non-metric multidimensional scaling (NMDS) ordination based on quantitative Bray–Curtis dissimilarities of relative abundance of bleaching categories was used to visualise bleaching impact across depth and the other tested factors. Significant differences in bleaching distribution were tested by permutational multivariate analysis of variance (PERMANOVA) using Bray–Curtis dissimilarity matrices after testing for the homogeneity of multivariate dispersions using PERMDISP. A similarity percentage test (SIMPER) was used to identify the bleaching categories responsible for most variation between tested treatments. The same ordination and hypotheses-based multivariate statistics were applied to analyse variation in coral community structure (with genus resolution) over depth.

Univariate OLR was used to model the contribution of the explanatory factors depth, location, region (Far Northern versus Ribbon Reefs) and coral taxonomy to the level of bleaching of individual colonies for $n = 3394$ colonies representing the aforementioned common taxa (12 genera). OLR is an extension of binomial logistic regression in which multiple categories associated with an ordinal event are

attributed an odds ratio statistic—the probability of the occurrence of that event based on a combination of one or more independent variables. This approach allows keeping the information contained in the ordering of bleaching categories and thus ascertaining the probability of bleaching progression (from healthy to recently dead). Inclusion of explanatory variables and their interactions in the model was determined by a forward and backward model selection tool based on the Akaike Information Criteria. Proportion of explained variation (pseudo-$R^2$) by the model was determined using McFadden's $R^2$, which compares the log-likelihood values of the fitted model against those of a null model containing only the intercept. The same model comparison approach was applied to calculate proportion of explained variation by each of the explanatory variables. For the calculation of the odd ratios, the categories "Far Northern" and "*Acropora*" were used as reference for the factors region and coral taxonomy, respectively. Interaction was conceptualised in terms of log odds (logits) and their significance (at $p < 0.05$) against the log odds of reference categories assessed using a Student's *t*-test. The OLR model (main variables and interaction effects) was then validated by a nested model approach using Chi-square statistics. All statistics were performed in R version 3.4.0[39] using packages "vegan"[40], "foreign"[41], "ggplot2"[42], "MASS"[43], "Hmisc"[44], and "reshape2"[45].

**Data availability**. Raw temperature data used in this manuscript (and future collected data) can be downloaded from https://github.com/pimbongaerts/monitoring. All other relevant data are available from the authors.

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

## Acknowledgements

We thank David Whillas and Luke Densely for their assistance with the ROV operations, and Kyra Hay, Jaap Barendrecht, Paul Muir and the crews from "Reef Connections", "Mike Ball Dive Expeditions", "SY Ethereal" and the "Waitt Foundation/MV Plan B" for support in the field. We like to thank the staff of the Global Change Institute (in particular David Harris and Sara Naylor) and the Ocean Agency for their logistical support. We thank Tobias Robinson for performing the logger calibrations at the Australian Institute of Marine Science. We thank Paul Muir for his comments on a previous version of the manuscript. We thank the XL Catlin Seaview Survey for funding, as well as Waitt Foundation and Shannon and Bill Joy for their generous support of the project. The temperature monitoring study was undertaken as part of the XL Catlin Seaview Survey, designed and undertaken by the Global Change Institute, and funded by the XL Catlin Group in partnership with Underwater Earth and The University of Queensland. The bleaching surveys were undertaken as part of an ARC Discovery Early Career Researcher Award (DECRA) DE160101433. Substantial sea-time was generously provided on the "MV Plan B" by the Waitt Foundation and on the "SY Ethereal" by the Joy Foundation. P.R.F. was supported by the Portuguese Science and Technology Foundation (FCT) through fellowship SFRH/BPD/110285/2015. O.H.G. was supported by an Australian Research Council (ARC) Laureate Fellowship and was Deputy Director of the ARC Centre for Excellence in Coral Reef Studies during the project.

## Author contributions

P.B., N.E. and O.H.G. designed the study; P.R.F., P.B., N.E. and M.G.R. contributed to data collection; N.E. conducted the video analysis; P.R.F., N.E. and A.R. performed data processing and produced figures; P.R.F. conducted the data analysis; P.R.F., P.B. and N.E. wrote the manuscript, with all authors contributing to manuscript revisions; P.B. and O.H.G. attracted funds for the project.

## Additional information

**Competing interests:** The authors declare no competing interests.

