## [Peer Review File · Nature Communications]

Reviewers' comments:

Reviewer #1 (Remarks to the Author):

General Comments:

The study is an interesting characterization of shallow (5m) to deep (100m) thermal regimes before the 2016 Great Barrier Reef bleaching event and during the event for temperatures and corals (to 40m). The major conclusion are that there was some buffering of deeper reef communities, but the buffering broke down after the summer, and there was taxa-specific bleaching and mortality to 40m. The results are well presented and compelling. The novelty of the manuscript seems to be the first characterization of the bleaching response of GBR mesophotic reefs. While this is important and will be of interest to coral reef researchers, I am not sure it will speak to a broader audience. The bleaching response of deeper and mesophotic corals has been published before in reference to the deep reef refugia hypothesis (Lang et al. 1988; Bunkley-Williams et al. 1991; Smith et al. 2016, Muir et al. 2018). This is in no way a reflection of the quality of the paper, but in my opinion the study may not have sufficiently broad appeal for Nature Climate Change.

Specific Comments:

Abstract:

Lin 22-23. That the study evaluated the potential of 40-100m reefs is a bit of a stretch. No deeper temperature records below 40m extended into the 2016 event and no bleaching observations were made below 40 m.

Figure 1. This is a little hard to read on the pdf. Perhaps it will look okay in the final version, but you might consider larger text.

Line 77. "least affected"

Lines 73-82. This seems like a lot of confirmatory methods to be in a figure caption. I would suggest moving to the methods.

Figure 2. It would be helpful to indicate when a temperature record at a particular depth ceased. E.g., at Osprey Bigeye Ledge Does the 60m record stop in Jan 2015 or is it under the 40m data?

Line 111-112. "suggesting different thermal stress thresholds over depth" I see where you are going but I think this statement needs to be developed a little more because I don't think it will be apparent to general readers.

Supplemental Figure 1. The depths of the panels are in the text, but it would be helpful to have the depths on the panels themselves. There is plenty of room.

Lines 124-156 and throughout the text. The use of round brackets is fine, but it gets a bit excessive and unnecessary at times. It make it seem like there is a second aside conversation going on in the text. My personal preference would be to have more of the bracketed material as part of the sentence and separated with commas where necessary.

Line 287-289. Smith et al. (2016) [reference 13 in this paper] also showed delayed onset and longer duration of heat stress in mesophotic reefs. This would be an appropriate citation to show that this phenomena may be at work in at least two ocean basins.

Methods

Line 393. "inquire" is an awkward word choice.

Supplemental Material

Line 68-70. May want to mention Suppl. Fig. 3 with this statement, since it shows the genera-specific responses and relates to this statement.

Supplementary Figure 4. Include more details on the photograph locations and depths. Would also be nice to point out the major species shown as bleached. Do you need to include photo credits?

Reviewer #2 (Remarks to the Author):

This manuscript is a useful contribution to the literature. We have waited for some time to get information on deep reefs during thermal stress events, especially from the Great Barrier Reef that suffered mass bleaching in 2016 and 2017. Although the science is sound, the manuscript needs some serious editing.

Major concerns

Most of the figure captions have extensive sections that discuss the data. Discussions are inappropriate in figure captions. Rewrite all figure captions, and shorten them all, particularly Figures 2, 3 and 4.

There are way too many sentences starting with "This, ...", without qualifying what this is referring to.

Page 11. The authors switch back and forth from shallow to deep sites. This is the most important part of the text and the reader gets confused which depth the authors are discussing. This entire page needs careful thought and rewriting. For example, Line 208. It is unclear which sites the authors are referring to, is it the shallow or deep sites. Best to say, At our 40 m GBR sites. It is unclear which species are most affected at the shallow and which are most affected at the deep sites. A table comparing shallow and deep coral sensitivities would help greatly.

Edward Tuft, the statistician, wrote a book a few years back about the best and worst ways to present scientific data. He stated that the absolute worst way to present scientific data was using pie diagrams. Unfortunately, most people working on endosymbionts use pie diagrams, and in the present manuscript Figures 1 and 4 are a series of pie diagrams. There must be a better way to present these data.

There are lots of studies on shallow water coral susceptibility to thermal stress that the authors have missed, including some classic studies that also discuss the susceptibility of *Seriatopora* and *Stylophora*. The present authors are not the first to mention this susceptibility.

The present manuscript needs a table that compares the susceptibilities between corals in shallow and deep systems. Are the deep systems merely a dampened version of the shallow systems? Or, is the present work suggesting something else? The comparative temperature profiles are clear, but the comparisons of the coral responses between depths are not clear, at least as currently presented.

In the methods it is unclear how many benthic video transects were used at each site? Hope it was not just one per site.

Page 19. Paragraph 2. The authors refer to 4 different multivariate techniques, without really justifying them. It would be best to outline what you intend to do first, and then mention the tools you used to do it. Are all 4 methods really necessary for this study.

Minor concerns

Line 22. Empirical assessments are limited, not remain limited.

Line 23, (40-100 m) should follow when mesophotic is first mentioned in line 20.

Line 26. "colder temperatures" . It is unclear what cooler is being compared with; colder than what? This is the Abstract. Yet there is no context. There is no mention of temperatures until line 29. Rewrite.

Line 31. Rerword as follows: ...lower than shallower depths...

Line 34. What does this mean: "the temporal window needed for this relief"? Rewrite.

Line 42. ... mesophotic coral ecosystems ... should not be in quotation marks. Be consistent. Is it > 30 m or 40 m as in the Abstract. What is the depth range?

Line 43. What is "This is,..." Referring to?

Line 49. ...and to facilitate the recovery..." (not potentiate)

Line 50. Should read ... mass coral bleaching...

Line 52. Should be an en-dash, not a hyphen for 2015 to 2016.

Line 53. "...in the far northern..." (not in its far northern)

Line 58 to 60. Rewrite. What is "This" referring to?

Line 66. There is no a and b in the figure, even though the figure caption refers to a and b.

Line 77. Less should be least.

Line 82. March/April should be March – April.

Line 99. ... compared with (not compared to).

Line 110. .. compared with (not compared to)

Line 139. Colder than what?

Line 145. These maximum DHW values at 40 m were ...

Line 149. ... compared with

Line 154. What is "it" referring to?

Lines 164, 171, 187, 190. ... compared with (not compared to)

Lines 195 and 196. Awkward. Rewrite.

Line 198. Rewrite. You are not interrogating, but rather partitioning the variance.

Lines 199 to 202. Rewrite this section. Extremely awkward.

Line 206. increase in, not increase on.

Line 215. Rewrite. What is a future abundance distribution. You can have distributions and abundances, but not both together, unless this terminology is clearly explained.

Lines 226 to 238. Delete from figure caption and incorporate in text. There are lots of studies on shallow water coral susceptibility to thermal stress that the authors have missed, including some classic studies in 2001 and 2002, which also discuss the susceptibility of *Seriatopora* and *Stylophora*.

Figure 5. It is unclear what the differences are between the left and the right-hand figures. The authors state that b is an interaction plot with depth, but depth is a covariate in the figure. This can't be correct if you are comparing the other covariates with depth (you can't compare depth to itself). This figure needs to be clearly explained. Again, the figure caption is too long and includes discussion.

Line 276. If you mention a major current, such as the Hiri Current, this current should be shown in Figure 1.

Line 289. In the onset of what?

Line 315. Totalizing is an English word, but rarely used. Rewrite.

Line 328. Use commas between exposures, at 10 ...bleaching year, ...

Line 329. Compared with.

Lines 339 to 340. Awkward. Rewrite.

Line 361. How many benthic video transects were used at each site?

Lines 375 to 385. This eight line sentence needs to be partitioned. Probably best to do so after "...supplementary material). Secondly....

Line 328. Rewrite. Also replace totalizing.

Lines 410 to 415. Awkward English. Rewrite.

Line 426. Refer to the R package version and the authors that wrote the packages.

Reviewer #3 (Remarks to the Author):

This paper tests the potential of mesophytic reefs on the GBR and Western Coral Sea to serve as refugia for heat stress. The work appears to adequately test this, although I recommend adding another supplemental figure directly comparing community composition with depth and location to help clarify the relations among these variables. My biggest concern in the analysis is the pooling of data across sites when it appears there may be significant differences. The validity of this pooling needs to be demonstrated.

Additionally, the authors need to address a philosophical question: is a mesophytic reef a refuge if it contains a different set of species than the shallow reef? It seems that if the taxa are fundamentally different, most of the mesophytic reef is not serving as a refuge.

Once these issues are resolve, I believe this paper will be a significant contribution toward understanding an important question that is becoming more critical as accelerating climate change continues to degrade coral reefs around the world.

Reviewer #1	Reply to comments of Reviewer #1
Remarks to the Author: The study is an interesting characterization of shallow (5m) to deep (100m) thermal regimes before the 2016 Great Barrier Reef bleaching event and during the event for temperatures and corals (to 40m). The major conclusion are that there was some buffering of deeper reef communities, but the buffering broke down after the summer, and there was taxa-specific bleaching and mortality to 40m. The results are well presented and compelling. The novelty of the manuscript seems to be the first characterization of the bleaching response of GBR mesophotic reefs. I would recommend publication of this manuscript after minor revisions.	✔ We are very pleased with the reviewer’s general comment and committed to incorporate all the specific comments raised below. In our responses we refer to the line numbers where the respective changes can be found in the manuscript file with highlighted track changes.
Specific Comments: Abstract: Line 22-23. That the study evaluated the potential of 40-100m reefs is a bit of a stretch. No deeper temperature records below 40m extended into the 2016 event and no bleaching observations were made below 40 m.	✔ We agree this statement was a bit of a stretch and have removed the depth range from the sentence. The revised sentence reads: “We evaluated the potential of mesophotic reefs within the Great Barrier Reef (GBR) and adjacent Coral Sea to act as thermal refuges by characterizing long-term temperature conditions and assessing impacts during the 2016 mass bleaching event”. Please see lines 22-25.
Figure 1. This is a little hard to read on the pdf. Perhaps it will look okay in the final version, but you might consider larger text.	▲ We appreciate the reviewer’s comment and will liaise with NCOMMS about that issue, as in our version it is still readable.
Line 77. “least affected”	✔ Corrected. Please see line 484.
Lines 73-82. This seems like a lot of confirmatory methods to be in a figure caption. I would suggest moving to the methods.	✔ We agree with the reviewer’s assessment that it should be moved to the methods. It is now to be found in lines 481-489.
Figure 2. It would be helpful to indicate when a temperature record at a particular depth ceased. E.g., at Osprey Bigeye Ledge Does the 60m record stop in Jan 2015 or is it under the 40m data?	✔ We agree with the reviewer that it would be helpful - and have therefore added these details to the figure caption: “On the GBR sites all recordings for 60–100 m ceased in March 2014. At the Osprey sites in the WCS the 60 m recordings ceased in January 2015, and for Holmes the 10 and 60 m loggers stopped recording in November 2013, whereas the 40 m logger went on recording until February 2015”. Please see lines 761-765.
Line 111-112. “suggesting different thermal stress thresholds over depth” I see where you are going but I think this statement needs to be developed a little more because I don’t think it will be apparent to general readers.	✔ We totally agree with the reviewer and have now expanded to further explain and develop this idea. The new sentence reads: “These differential long-term patterns of exposure of different depths to higher and lower temperatures suggest different thermal stress thresholds for corals living at different depths (as physiological tolerances are often adjusted to prevailing conditions)“. Please see lines 126-129.
Supplemental Figure 1. The depths of the panels are in the text, but it would be helpful to have the depths on the panels themselves. There is plenty of room.	✔ We agree - thank you for the suggestion. Suggestion incorporated into Suppl. Fig. 1. Please see Supplementary Material.
Lines 124-156 and throughout the text. The use of round brackets is fine, but it gets a bit excessive and	✔ We followed the suggestion of the reviewer and have given preference to the use of commas where

unnecessary at times. It make it seem like there is a second aside conversation going on in the text. My personal preference would be to have more of the bracketed material as part of the sentence and separated with commas where necessary.	appropriate. Please see lines 137, 155, 194 and 195.
Line 287-289. Smith et al. (2016) [reference 13 in this paper] also showed delayed onset and longer duration of heat stress in mesophotic reefs. This would be an appropriate citation to show that this phenomena may be at work in at least two ocean basins.	✔ We are grateful to the reviewer for pointing this out and have added this citation to make the stronger case that this phenomenon may be at work in two distinct ocean basins. We have added this new sentence: “Such a delayed onset and longer duration of heat stress in mesophotic reefs than in their shallow water counterparts has previously been shown for the Caribbean region“. Please see lines 357-359.
Methods Line 393. “inquire” is an awkward word choice.	✔ Replaced with “investigate”. See line 514.
Supplemental Material Line 68-70. May want to mention Suppl. Fig. 3 with this statement, since it shows the genera-specific responses and relates to this statement.	✔ Suggestion incorporated. See line 78.
Supplementary Figure 4. Include more details on the photograph locations and depths. Would also be nice to point out the major species shown as bleached. Do you need to include photo credits?	✔ Done as requested. New caption reads: “Photo examples: a) Ribbon Reef #10 at 35 m depth with many severely bleached coral colonies including Stylophora sp., Montipora sp. and Seriatopora sp., b) transect method used to score bleaching impact, note a distance of 1m between the two poles protruding out of the video frame held by the diver (photo taken on Ribbon Reef #10 at 25 m depth) , and c) video still of shallow bleaching (10 m depth) within transect (with the poles marking the 1m-wide swath). All photographs by Pim Bongaerts and licensed under CC BY-SA 4.0“. Please see lines 122-128.

Reviewer #2	Reply to comments of Reviewer #2
Remarks to the Author: This manuscript is a useful contribution to the literature. We have waited for some time to get information on deep reefs during thermal stress events, especially from the Great Barrier Reef that suffered mass bleaching in 2016 and 2017. Although the science is sound, the manuscript needs some serious editing.	✔ We agree with the reviewer that this contribution is timely, particularly given the extent of speculation on the potential role of deeper reef areas in the media. We are grateful for the style concerns raised, and we have incorporated practically all the suggestions. In our responses we refer to the line numbers where the respective changes can be found in the manuscript file with highlighted track changes.
Major concerns Most of the figure captions have extensive sections that discuss the data. Discussions are inappropriate in figure captions. Rewrite all figure captions, and shorten them all, particularly Figures 2, 3 and 4.	✔ We have now deleted or moved all of these sections that discuss the data into the manuscript body, and we have also considerably shortened the captions. Please see lines 757-797, 803-814, and 840-855 for new captions.
There are way too many sentences starting with “This, ...”, without qualifying what this is referring to.	✔ Thank you for the suggestion, we have greatly reduced the use.
Page 11. The authors switch back and forth from shallow to deep sites. This is the most important part of the text and the reader gets confused which depth the authors are discussing. This entire page needs careful thought and rewriting. For example, Line 208. It is unclear which sites the authors are referring to, is it the shallow or deep sites. Best to say, At our 40 m GBR sites. It is unclear which species are most affected at the shallow and which are most affected at the deep sites. A table comparing shallow and deep coral sensitivities would help greatly.	✔ We appreciate the reviewer’s comment and did our best to incorporate it and make the text more clear. So, when no depth is mentioned this means we are referring to a general phenomenon transversal to depth. We tried to now make that explicit, when needed. See line 242, for example. With regard to “which species are most affected at the shallow and which are most affected at the deep sites”... our approach was to “disentangle the contribution of changes in community structure from that of depth to the bleaching impact”, as mentioned. As such, we refer to the susceptibility of coral genera irrespective of their depth range, and we refer to the general effect of depth. However, Fig. 5 and Suppl. Fig 4 do get in greater detail about which coral genera experience strong depth effects on their bleaching response, and which ones show no evident effect of depth. We have therefore brought some of that information into the body of the manuscript. Please see lines 258-284 reading: “On top of that, some bleaching-sensitive taxa seem to experience a strong effect of depth on their bleaching response (i.e., Pachyseris, Dipsastraea and Seriatopora), as confirmed by the odds ratios for the interaction of specific taxa with depth (Fig. 5b). Identification of “bleaching-tolerant” taxa, “bleaching-sensitive” taxa and highly sensitive taxa with “dampened response over depth” clearly reflect known taxa-specific patterns of bleaching susceptibility versus tolerance often attributed to physiological properties inherent to the coral animal itself or to its symbiotic communities”. ▲ We do want to refrain from taking definite conclusions on the response of individual coral species since we have only identified corals to the genus level and there is a broad literature that already covers the topic of bleaching susceptibility of distinct coral species. The consensus is that even within each coral genus there may be lineages that are more or less susceptible to bleaching. Therefore, we decided not to add such table as suggested by the reviewer because

	this information could be misleading, but we did reorganize Fig. 4 according to the order of taxa susceptibility to bleaching as modeled by the OLR model: “bleaching-tolerant” taxa, “bleaching-sensitive” taxa and highly sensitive taxa with “dampened response over depth”. Please see this new version of Fig. 4.
Edward Tuft, the statistician, wrote a book a few years back about the best and worst ways to present scientific data. He stated that the absolute worst way to present scientific data was using pie diagrams. Unfortunately, most people working on endosymbionts use pie diagrams, and in the present manuscript Figures 1 and 4 are a series of pie diagrams. There must be a better way to present these data.	▲ We obviously respect the opinion of the reviewer and realize that the use of pie charts is a contentious topic. The main criticism with pie graphs is that angles are intuitively harder to compare, which is an issue when differences are small and many categories are used. We believe this issue does not really apply to our data (particularly given that there are substantial differences and sequential categories), and that pie graphs are actually well-suited in a grid-layout which allows for easy comparison in two dimensions (in this case: location and depth). Nonetheless, we are happy to change to stacked bars and have therefore updated the figure accordingly. Please see Figs. 1 and 4.
There are lots of studies on shallow water coral susceptibility to thermal stress that the authors have missed, including some classic studies that also discuss the susceptibility of Seriatopora and Stylophora. The present authors are not the first to mention this susceptibility.	✔ We agree with the reviewer and want to clarify that we had no intention to state that we were the first to present on that. We now add a reference to a classical study on the susceptibility of Seriatopora and Stylophora. Please see line 300 and the new reference in the literature: “Hoegh-Guldberg, O. & Smith, G. J. The effect of sudden changes in temperature, light and salinity on the population density and export of zooxanthellae from the reef corals Stylophora pistillata Esper and Seriatopora hystrix Dana. J. Exp. Mar. Biol. Ecol. 129, 279-303 (1989)“.
The present manuscript needs a table that compares the susceptibilities between corals in shallow and deep systems. Are the deep systems merely a dampened version of the shallow systems? Or, is the present work suggesting something else? The comparative temperature profiles are clear, but the comparisons of the coral responses between depths are not clear, at least as currently presented.	▲ At the moment we do not see the need to add such a table as Fig. 5 does cover both the depth effect, the response of the distinct corals and the interaction effects of depth for each taxon. However, our updated Fig. 4 now groups the distinct taxa according to their susceptibilities during the 2016 bleaching event. Furthermore, we now also provide an overview on how the relative abundances of bleaching-tolerant and bleaching-sensitive taxa change with depth (new Suppl. Fig. 3), as a means to clarify whether the reduction of bleaching incidence with depth is a consequence of a change in community composition to more tolerant taxa, or whether there is evidence for thermal relief. This also led to some new sections in the text. See lines 261-289: “Identification of “bleaching-tolerant” taxa, “bleaching-sensitive” taxa and highly sensitive taxa with “dampened response over depth” clearly reflect known taxa-specific patterns of bleaching susceptibility versus tolerance often attributed to physiological properties inherent to the coral animal itself or to its symbiotic communities. The observed greater proportions of thermally-tolerant coral taxa present on the shallow reef and that of highly sensitive genera on the deeper reef (Suppl Fig. 3b), suggest that the relief in bleaching incidence offered by the deep reef cannot be solely explained by differences in community composition

	and that there is at least some degree of thermal relief with depth". We hope this new material meets the reviewer's concerns about "the comparisons of the coral responses between depths" and helps answering the interesting questions brought up by the reviewer. We want to state that we share the same concerns. In fact, the last 8 lines of our manuscript focus exactly on this, and our conclusion is that "the lower incidence of bleaching that was observed on the deep reef coincides with a considerable shift in the community structure. This indicates that the proportion of shallow-water diversity protected in sufficiently large numbers at depth may be limited, and that a subsequent role as reproductive source aiding in shallow reef recovery may be restricted to a relatively small proportion of species". Please see lines 359-373.
In the methods it is unclear how many benthic video transects were used at each site? Hope it was not just one per site.	▲ It was in fact one long, continuous transect of 75 m in length. However, we utilize an Ordinal Logistic Regression statistical approach dealing with individual entries (by scored colonies rather than counts across transects). Ideally, we would have done multiple of those transects at each site, however given the very limited bottom time at depth (due to the strict scientific diving regulations in Australia) and limited time at sea, we decided it would be better to prioritize more sites, rather than transect replication within sites. However, comparisons amongst sites show the data is robust, particularly when looking at sites located close to each other (e.g., sites Mantis #1–4). Statistical tests showed no differences between individual sites, but put the emphasis on differences between two larger regions. This is also now made clearer in lines 906-908: "Location had no significant effect on the progression of bleaching as tested by OLR and individual locations were therefore grouped into two regions".
Page 19. Paragraph 2. The authors refer to 4 different multivariate techniques, without really justifying them. It would be best to outline what you intend to do first, and then mention the tools you used to do it. Are all 4 methods really necessary for this study.	▲ We think that although the methods are not all necessary to support the main findings (which are instead based on the OLR approach), they provide a good validation of the main method. In fact, finding the same trends by using distinct methods shows the robustness of trends. Since these other methods are only shown in the supplementary and since OLR is not a commonly used statistical approach we would prefer to keep the reference to these other methods and respective results in the supplemental material.
Minor concerns Line 22. Empirical assessments are limited, not remain limited.	✔ Corrected as suggested. Please see line 22.
Line 23, (40-100 m) should follow when mesophotic is first mentioned in line 20.	▲ We did not mean to define the depth range of mesophotic reefs here (we do so in the beginning of the introduction), but to actually give the depth range of our study. We have therefore not followed the suggestion.
Line 26. "colder temperatures". It is unclear what cooler is being compared with; colder than what? This	✔ Due to space restrictions the abstract has been partially re-written and we have therefore removed

is the Abstract. Yet there is no context. There is no mention of temperatures until line 29. Rewrite.	these unclear sections. Please see lines 25-27.
Line 31. Reword as follows: ...lower than shallower depths...	✔ Suggestion accepted. Please see line 29.
Line 34. What does this mean: “the temporal window needed for this relief”? Rewrite.	✔ We rephrased this to " transient nature of the protection " - although it would be great to expand and explain further we had to cut down the abstract to ca. 150 words and there simply is no room. Nonetheless, we have used the same terminology in the discussion where we provide a clear explanation (i.e. refuge can only be provided in summer when upwelling occurs). Please see line 30-32.
Line 42. ... mesophotic coral ecosystems ... should not be in quotation marks. Be consistent. Is it > 30 m or 40 m as in the Abstract. What is the depth range?	✔ We have removed the quotation marks. Please see lines 39. This question has been addressed earlier (i.e. study depth range vs "mesophotic" depth range).
Line 43. What is “This is,....” Referring to?	✔ It is referring to the poor evaluation so far given to the potential for deeper sections of coral reefs to provide a refuge against thermal bleaching. We have rephrased to improve clarity. Please see lines 40-41.
Line 49. ...and to facilitate the recovery...” (not potentiate)	✔ Corrected as suggested. Please see line 45.
Line 50. Should read ... mass coral bleaching...	✔ Corrected as suggested. Please see lines 73-74.
Line 52. Should be an en-dash, not a hyphen for 2015 to 2016.	✔ Corrected as suggested. Please see line 75.
Line 53. “...in the far northern...” (not in its far northern)	✔ Corrected as suggested. Please see line 76.
Line 58 to 60. Rewrite. What is “This” referring to?	✔ It refers to the situation detailed in the previous sentence, that the bleaching impacts at mesophotic depths remain unknown. Please see lines 81-83 for rewritten version: “ Despite this lack of knowledge, mesophotic reefs are estimated to represent a surface area equivalent to that of shallow reefs on the GBR... ”.
Line 66. There is no a and b in the figure, even though the figure caption refers to a and b.	✔ Done. Please see new Fig. 1.
Line 77. Less should be least.	✔ Corrected. Please see line 484.
Line 82. March/April should be March – April.	✔ Corrected. Please see line 489.
Line 99. ... compared with (not compared to).	✔ Corrected. Please see line 122.
Line 110. ... compared with (not compared to)	✔ Corrected. Please see line 124.
Line 139. Colder than what?	✔ Changed to "cold-water influxes". Please see line 161.
Line 145. These maximum DHW values at 40 m were...	✔ Corrected as suggested. Please see lines 178-179.

Line 149. ... compared with	✔ Corrected. Please see line 182.
Line 154. What is “it” referring to?	✔ It refers to the Osprey temperature data. Corrected to clarify. Please see lines 192-194: “ Regardless, our Osprey temperature data demonstrates that oceanographic settings with stronger upwelling patterns... ”.
Lines 164, 171, 187, 190. ... compared with (not compared to)	✔ Corrected. Please see lines 805-806, 167, 204, 207, 246, 248 and 338.
Lines 195 and 196. Awkward. Rewrite.	✔ We have rewritten this sentence. Please see lines 212-213: “ In order to disentangle the contributions of coral community structure and depth to the bleaching impact ”
Line 198. Rewrite. You are not interrogating, but rather partitioning the variance.	✔ We have replaced “interrogate” with “ascertain”. Please see line 215.
Lines 199 to 202. Rewrite this section. Extremely awkward.	✔ Section has been rewritten. Please see lines 234-238: “ OLR confirmed that with increasing depth, the chance of bleaching and bleaching-related mortality decreased slightly though significantly. For a one-unit increase in depth, one should expect about 4.8% decrease (i.e., 0.952 increase; see odds ratio for depth in Fig. 5a) in the odds of being in a higher bleaching category (if holding region and coral taxa at a fixed value) ”.
Line 206. increase in, not increase on.	✔ Replaced with “ increase in ”. Please see line 244.
Line 215. Rewrite. What is a future abundance distribution. You can have distributions and abundances, but not both together, unless this terminology is clearly explained.	✔ Abundance distribution is the distribution of abundances, for instance, across an environmental gradient. To clarify, we have now separated “distributions” and “abundances”. The new sentence reads: “ will determine their future abundances and distribution ”. Please see line 296-297.
Lines 226 to 238. Delete from figure caption and incorporate in text. There are lots of studies on shallow water coral susceptibility to thermal stress that the authors have missed, including some classic studies in 2001 and 2002, which also discuss the susceptibility of Seriatopora and Stylophora.	✔ Done as requested. Please see incorporation of this section into lines 261-303. The comment about the existence of further studies on the susceptibility of corals to thermal stress has been addressed above.
Figure 5. It is unclear what the differences are between the left and the right-hand figures. The authors state that b is an interaction plot with depth, but depth is a covariate in the figure. This can’t be correct if you are comparing the other covariates with depth (you can’t compare depth to itself). This figure needs to be clearly explained. Again, the figure caption is too long and includes discussion.	✔ Thank you for this feedback. The differences between the two panels are explained in the figure caption, where the left panel shows the explanatory variables and the right panel their interactions with depth. We have modified the caption to improve on the explanation, while shortening by removing some of the examples. We have also removed the variable depth from the right panel as we agree this was indeed confusing. Please see lines 893-933.
Line 276. If you mention a major current, such as the Hiri Current, this current should be shown in Figure 1.	▲ We appreciate the reviewer’s opinion but in this case decided not to accept the suggestion as this would implicate the Hiri Current has a major role in the story.
Line 289. In the onset of what?	✔ Clarified. New sentence reads: “ Due to the lag in ”

	the onset of elevated temperatures at mesophotic depths... ". Please see lines 354-355.
Line 315. Totalizing is an English word, but rarely used. Rewrite.	✔ Replaced by "totaling". Please see line 403.
Line 328. Use commas between exposures, at 10 ...bleaching year, ...	✔ Done as suggested. Please see lines 429-430.
Line 329. Compared with.	✔ Corrected. Please see line 430.
Lines 339 to 340. Awkward. Rewrite.	✔ We have broken that sentence in two and rewritten. It now reads: " This approach allows estimating heat stress thresholds for corals that live at mesophotic depths and whose acclimatization history has not been considered so far. This was done by assuming that their bleaching response to temperature is based on the same premises as is well-established for shallow-water corals ". Please see lines 440-444.
Line 361. How many benthic video transects were used at each site?	▲ Similar as to above - we considered all the individually scored colonies within a 75x1m swath (i.e. belt transect) at each depth (so our statistical unit is the individual colonies rather than transects).
Lines 375 to 385. This eight line sentence needs to be partitioned. Probably best to do so after "...supplementary material). Secondly....	✔ Done as suggested. Please see line 501.
Line 328. Rewrite. Also replace totalizing.	✔ Rewritten. Please see line 504-506.
Lines 410 to 415. Awkward English. Rewrite.	✔ We have provided some minor changes and hope the English is not awkward any longer. Please see lines 536-538.
Line 426. Refer to the R package version and the authors that wrote the packages.	✔ Done. Please see line 554 and the new entry in the literature references.

Reviewer #3	Reply to comments of Reviewer #3
Remarks to the Author: This paper tests the potential of mesophytic reefs on the GBR and Western Coral Sea to serve as refugia for heat stress. The work appears to adequately test this, although I recommend adding another supplemental figure directly comparing community composition with depth and location to help clarify the relations among these variables.	✔ We thank the reviewer for the useful comments. In our responses we refer to the line numbers where the respective changes can be found in the manuscript file with highlighted track changes. We have added the suggested (supplementary) figure (new Suppl. Fig. 3) to highlight the change in community composition with depth and location, and have made changes to address the remaining concerns listed below.
My biggest concern in the analysis is the pooling of data across sites when it appears there may be significant differences. The validity of this pooling needs to be demonstrated.	✔ We apologize for this confusion, but in fact site/location was included in our statistical tests. However, the model selection tool we used for our multivariate approach did consistently pick region, coral taxa and depth, but not location, as the best factors to explain the variability in the dataset. Site location was actually not a significant factor, and this was also the case for our OLR statistics. We believe that with a higher number of observations per location, this factor could have maybe become significant as well, but the reality is that “region” (which combines the different sites into two levels) provides the only significant explanatory variable for the geographic variation in the system. We have now included a clarification to the caption of supplementary Fig. 2: “Unlike region, individual site/location had no significant effect on the distribution of the distinct bleaching categories” (please see lines 50-51 of supplemental material), and to caption of Fig. 5: “Location had no significant effect on the progression of bleaching as tested by OLR and individual locations were therefore grouped into two regions” (please see lines 906-908).
Additionally, the authors need to address a philosophical question: is a mesophytic reef a refuge if it contains a different set of species than the shallow reef? It seems that if the taxa are fundamentally different, most of the mesophytic reef is not serving as a refuge.	✔ This is similar to a comment from Reviewer #2. We responded that we completely agree with this assessment, and the last section of our manuscript focuses on this discussion. We conclude that “the lower incidence of bleaching that was observed on the deep reef coincides with a considerable shift in the community structure. This indicates that the proportion of shallow-water diversity protected in sufficiently large numbers at depth may be limited, and that a subsequent role as reproductive source aiding in shallow reef recovery may be restricted to a relatively small proportion of species”. Please see lines 359-373.
Once these issues are resolve, I believe this paper will be a significant contribution toward understanding an important question that is becoming more critical as accelerating climate change continues to degrade coral reefs around the world.	✔ We are appreciative of these supporting words.
The bleaching response of deeper and mesophotic corals has been published before in reference to the deep reef refugia hypothesis (Lang et al. 1988; Bunkley-Williams et al. 1991; Smith et al. 2016, Muir et al. 2018).	✔ We completely agree, and three of these references were already included. We now included "Bunkley-Williams et al. 1991" too. Please see line 43 and the new entry in the references: “Bunkley-Williams, L., Morelock, J. & Williams, E. H. Lingering Effects of the 1987 Mass Bleaching of Puerto Rican Coral Reefs in

	Mid to Late 1988. Journal of Aquatic Animal Health 3, 242-247, doi:10.1577/1548-8667(1991)003<0242:LEOTMB>2.3.CO;2 (1991) “.
Line 30 - It would be better if you include differences in levels of mortality relative to shallow reefs here.	▲ Normally we would agree, however given that surveys were done during the bleaching event, mortality alone is probably not a good indicator of disturbance impact.
Line 57 - It seems odd to speak of 2016 as both the event currently under study and a "past" event.	✔ Corrected by removing "past". Please see line 80.
Fig. 2 - It would help the reader to add GBR or WCS after the reef names. Please use colors other than red and green side by side. Many of us can't tell these apart. This becomes even more important in Figure 3. Fig. 3 - Please use colors other than red and green. Many of us can't tell these apart.	✔ Thank you for these useful suggestions - figure modified as suggested. All figures now optimized for color-blind readers.
Line 182 – “mass coral bleaching” and mortality	✔ Corrected: "mortality" included. Please see line 199.
Line 194 - While Figure 4 speaks to variations in bleaching sensitivity, it doesn't speak to the differential abundance with depth. I strongly recommend adding a figure comparing community composition across depths in the supplement. While it is possible that such information can be teased from Suppl. Figure 2, a straight-forward depiction of taxa vs depth would be very useful.	✔ We thank the reviewer for such relevant suggestion and we have now generated a new figure comparing community composition across depths. Please see new Suppl. Fig. 3a.
Line 216 - So a major issue here is the interaction between depth and taxa. You really need to more clearly address how much of the difference in bleaching with depth is due to (1) thermal stress exposure, or (2) changes in taxonomic composition.	✔ We thank the reviewer for bringing up this interesting issue. In order to address how much of the difference in bleaching with depth is due to changes in taxonomic composition we now present a new figure (Suppl. Fig. 3), following the reviewer's suggestion, but split into two parts/panels. First by using stacked bar graphs with the abundance of the 13 different taxonomic categories over depth (and location), and secondly by using 3 categories that concatenate the different taxa into “bleaching-tolerant”, “bleaching-sensitive” and “highly sensitive and dampened over depth”, as extracted from Fig. 5. This approach revealed that the observed lower incidence of bleaching on the deep reef coincides with a considerable shift in the community structure, with a greater proportion of thermally-tolerant coral genera present on the shallow reef and a greater proportion of genera highly sensitive to bleaching present on the deeper reef. This suggests that the difference in bleaching with depth cannot be solely explained by changes in community composition alone, and makes a stronger case for the role of thermal relief with depth. This led to a new section in the manuscript. Please see lines 261-289.
Fig. 4 - You should clearly indicate if n in Figure 4 is a valid representation of the depth effect on community composition (either here or in methods).	✔ We have now added this information to the caption of the Figure. This sentence reads: “ Number of observations noted next to each stacked bar are a valid representation of the depth effect on community composition (shown in Suppl. Fig. 3) ”. Please see lines 852-854.

Line 219 - Did you test for variance among locations within the GBR? Is it appropriate to combine data across locations?	✔ We did test for variance among locations and have clarified this in response to a previous comment by the reviewer.
Line 233 - Or could it be the other way around, that their survival at these depths could be related to past heat stress? How did the depth distribution vary from previously unbleached Northern GBR reefs to previously bleached Coral Sea reefs?	▲ Interesting idea - unfortunately we cannot evaluate this given that we did not have the opportunity (due to bad weather conditions) to survey any of the Coral Sea reefs. Although we feel it falls out of scope of this manuscript to discuss in detail patterns of coral species distribution over depth and their relation to particular susceptibilities of different taxa to bleaching, we would like to add to this interesting point of discussion that it is likely that the prevailing patterns of community composition over depth are, at the same time, a consequence of past heat stress and a determinant for future impacts of bleaching over depth. Species distributions reflect adaptation and selection in action and therefore reflect past selective pressures but will also determine the basis on which future selective pressures will act.
Line 249 - odds?	✔ Corrected. Please see line 900.
Line 252-261 - I find it hard to really tease the result you describe in the highlighted text from the data in the Figure. (a) has multiple variables, including depth, and (b) has interactions with depth. I think these figures and the text need a bit more explanation to help the reader sort the information out.	✔ Following a similar comment by reviewer 2, we have now modified the caption to improve on the explanation, while shortening by moving some specific examples to the main text. We have also removed the variable depth from the right panel as this was likely a cause for confusion, as reminded by reviewer 2. The main text has now been changed to provide a step-by-step interpretation of the figure, with reference to where in the figure this info can be found. Please see lines 214-254.
Line 275 - " when elevated temperatures extend ..."? "As" makes it sound like a regular occurrence.	✔ Suggestion accepted. Please see lines 340-341.
Line 283 - If the community composition is different, are they really a thermal refuge or just the home of more resistant species? Your "Secondly", below, really sounds like it is not truly a refuge.	✔ We have now added a new Fig in the supplementary depicting changes in the relative abundance of bleaching-tolerant vs bleaching-sensitive taxa. This new approach shows that, in general terms, the deeper reef does not just harbor more resistant species, and suggests that there is some reduction of thermal stress with depth. However, a role in shallow reef recovery may be restricted to a relatively small proportion of species that have broad depth distributions. Please see our concluding remarks in lines 370-375.
Line 294 - A refuge implies that the corals here can move up to the shallow reefs to supplement the shallow community at a time of loss. Is there evidence this is the case? It sounds more like the deep reef is the home to many species that can't/don't live in shallow reefs.	✔ Although the original definition of the "deep reef refuge hypothesis" included this "reseeding role", we address these concepts separately here, with refuge referring to a capacity to be buffered/protected from a disturbance (regardless of any subsequent "instrumental roles"). Regardless, our revised manuscript now includes a new figure prepared especially to answer these important questions. Please see replies above. As a summary, in general terms, we now show that the deeper reef is not just the home of more resistant species and this suggests that there is some reduction of thermal stress with depth. Please see discussion in lines 255-289: "At our GBR sites, coral

	taxa Porites, Leptoseris, Acropora and Pocillopora showed a comparatively low chance of experiencing bleaching and mortality, whereas taxa such as Stylophora, Isopora and Montipora were particularly prone to bleaching (see Fig. 4, genus-specific odds ratios in Fig. 5a, and Suppl. Fig. 4). On top of that, some bleaching-sensitive taxa seem to experience a strong effect of depth on their bleaching response (i.e., Pachyseris, Dipsastraea and Seriatopora), as confirmed by the odds ratios for the interaction of specific taxa with depth (Fig. 5b). Identification of “bleaching-tolerant” taxa, “bleaching-sensitive” taxa and highly sensitive taxa with “dampened response over depth” clearly reflect known taxa-specific patterns of bleaching susceptibility versus tolerance often attributed to physiological properties inherent to the coral animal itself or to its symbiotic communities. The observed greater proportions of thermally-tolerant coral taxa present on the shallow reef and that of highly sensitive genera on the deeper reef (Suppl Fig. 3b), suggest that the relief in bleaching incidence offered by the deep reef cannot be solely explained by differences in community composition and that there is at least some degree of thermal relief with depth”.
Line 384 - Why not just say totaling?	✔ Corrected. Please see line 505.
Line 393 - test? inquire into?	✔ Changed to "investigate". Please see line 514.

REVIEWERS' COMMENTS:

Reviewer #1 (Remarks to the Author):

The authors have done a commendable job in addressing my specific comments. I recommend publication.

Reviewer #2 (Remarks to the Author):

The manuscript has greatly improved; there are only a few minor issues to take care of before moving forward.

Line 28: reefs (not reef)

Line 34. ..because of (not due to)

Line 41. ...because of (not due to)

Line 106. This temperature anomaly, ... (not This, ...)

Line 213. Delete "All in all"

Line 214. Delete "the stage is set". Rewrite the sentence.

Line 248: Should read: ... a subsequent role of the mesophotic reefs as a reproductive source...

Line 291. Define DHW

Line 390. I insist that the authors of the R packages should be acknowledged and cited. These authors have spent a tremendous amount of time writing accessible code that they have made available to the world. It is standard practice and common courtesy to cite the authors of R packages.

The inclusion of Figure 4 has really improved the quality of the manuscript. Nice job.

Reviewer #3 (Remarks to the Author):

Thank you for the careful consideration of my earlier review comments. In general, I concur with your revisions. I especially appreciate the addition of figure S3a and the accompanying text that helped address the questions of community composition vs. response across depth. The clarification of these points is especially improved in the text such as that on page 8-9.

I think the following small items still deserves attention:

Lines 28 and 30. While I concur that mortality alone is not necessarily a good indicator, the combining of bleached and dead into a single value seems inappropriate. Rather than 46% bleached/dead at 40m and 70-81% bleached/dead at 5-25m, I recommend these numbers be broken out to read 40% bleached and 6% dead at 40m, and 60-69% bleached and 8-12% dead at 5-25m.

Line 169. Is this use of Author (date) citation appropriate in this journal?

Reviewer #1	Reply to comments of Reviewer #1
Remarks to the Author: The authors have done a commendable job in addressing my specific comments. I recommend publication.	We are very pleased with the previous comments provided by the reviewer and the way they contributed to improve our manuscript.

Reviewer #2	Reply to comments of Reviewer #2
Remarks to the Author: The manuscript has greatly improved; there are only a few minor issues to take care of before moving forward.	We are grateful for all the past and present comments by the reviewer. These have decisively contributed to improve our manuscript.
Line 28: reefs (not reef)	✔ Done as suggested. Please see line 28.
Line 34. ..because of (not due to)	✔ Done as suggested. Please see line 35.
Line 41. ...because of (not due to)	✔ Done as suggested. Please see line 42.
Line 106. This temperature anomaly, ... (not This, ...)	✔ Done as suggested. Please see line 121.
Line 213. Delete "All in all"	✔ Done as suggested. Please see line 251.
Line 214. Delete "the stage is set". Rewrite the sentence.	✔ This sentence was now re-written. It now reads "The mass bleaching event of 2016 has clearly demonstrated that, on top of the widespread coral mortality caused by a rapidly warming global climate, the fate of specific areas of the GBR and WCS is controlled by local oceanographic conditions". Please see lines 251-259.
Line 248: Should read: ... a subsequent role of the mesophotic reefs as a reproductive source...	✔ Done as suggested. Please see line 293.
Line 291. Define DHW	▲ We now give the full name of the DHW metric on the first occasion possible. Please see line 339. We already refer to a study offering a formal definition, so we do not consider we should include one. However, we explain in detail how this metric has been calculated in our study. Please see lines 339-342.
Line 390. I insist that the authors of the R packages should be acknowledged and cited. Theses authors have spent a tremendous amount of time writing accessible code that they have made available to the world. It is standard practice and common courtesy to cite the authors of R packages.	✔ Done as suggested. Six new references have therefore been added to the reference list. Please see lines 602-614.
The inclusion of Figure 4 has really improved the quality of the manuscript. Nice job.	We appreciate the comment!

Reviewer #3	Reply to comments of Reviewer #3
Remarks to the Author: Thank you for the careful consideration of my earlier review comments. In general, I concur with your revisions. I especially appreciate the addition of figure S3a and the accompanying text that helped address the questions of community composition vs. response across depth. The clarification of these points is especially improved in the text such as that on page 8-9.	We thank the reviewer for those previous comments that contributed to greatly improve the manuscript.
Lines 28 and 30. While I concur that mortality alone is not necessarily a good indicator, the combining of bleached and dead into a single value seems inappropriate. Rather than 46% bleached/dead at 40m and 70-81% bleached/dead at 5-25m, I recommend these numbers be broken out to read 40% bleached and 6% dead at 40m, and 60-69% bleached and 8-12% dead at 5-25m.	✔ Done as suggested. We note that we have slightly exceeded the maximum number of words allowed in the Abstract and leave it up to the editor to decide whether these changes are acceptable. Please see lines 28-30.
Line 169. Is this use of Author (date) citation appropriate in this journal?	✔ This has now been corrected to comply with the citation norms required by the journal. Please see line 193.